# *Peltigera* lichen thalli produce highly potent ice nucleating agents

Rosemary J. Eufemio[1,2], Galit Renzer[3], Mariah Rojas[4], Jolanta Miadlikowska[5], Todd L. Sformo[6], François Lutzoni[5], Boris A. Vinatzer[4], and Konrad Meister[1, 3*]

**Affiliations:**

[1]Department of Chemistry and Biochemistry, Boise State University, Boise, ID 83725, USA

[2]Biomolecular Sciences Graduate Programs, Boise State University, Boise, ID 83725, USA

[3]Max Planck Institute for Polymer Research, 55128 Mainz, Germany

[4]School of Plant and Environmental Sciences, Virginia Tech, Blacksburg, VA 24061, USA

[5]Department of Biology, Duke University, Durham, NC 27708, USA

[6]Institute of Arctic Biology, University of Alaska Fairbanks, Fairbanks, AK 99775, USA

*Correspondence to:* Konrad Meister (konradmeister@boisestate.edu)

**Abstract**

From extracellular freezing to cloud glaciation, the crystallization of water is ubiquitous and shapes life as we know it. Efficient biological ice nucleators (INs) are crucial for organism survival in cold environments and, when aerosolized, serve as a significant source of atmospheric ice nuclei. Several lichen species have been identified as potent INs capable of inducing freezing at high subzero temperatures. Despite their importance, the abundance and diversity of lichen INs are still not well understood. Here, we investigate ice nucleation activity in the cyanolichen-forming genus *Peltigera* from across a range of ecosystems in the Arctic, the Northwestern United States, and Central and South America. We find strong IN-activity in all tested *Peltigera* species, with ice nucleation temperatures above -12°C, and 35% of the samples initiating freezing at temperatures at or above -6.2°C. The *Peltigera* INs in aqueous extract appear resistant to freeze-thaw cycles, suggesting that they can survive dispersal through the atmosphere and thereby potentially influence precipitation patterns. An axenic fungal culture termed L01-tf-B03, from the lichen *Peltigera britannica* JNU22, displays an ice nucleation temperature of -5.6°C at 1 mg mL$^{-1}$ and retains remarkably high IN-activity at concentrations as low as 0.1 ng mL$^{-1}$. Our analysis suggests that the INs released from this fungus in culture are 1000 times more potent than the most active bacterial INs from *Pseudomonas syringae*. The global distribution of *Peltigera* lichens, in combination with the IN-activity, emphasizes their potential to act as powerful ice nucleating agents in the atmosphere.

**1 Introduction**

Ice formation below 0°C is thermodynamically favorable, however the crystallization process is constrained by kinetics. As a result, pure water droplets can be supercooled to temperatures as low as −38°C, below which homogeneous ice nucleation takes place (Koop et al., 2000). In natural systems, water typically freezes in a heterogeneous process facilitated by the presence of particles that serve as ice nucleators (INs). INs of biotic origin are often highly efficient and elevate freezing temperatures to -15°C and above (Maki and Willoughby, 1978; Wilson et al., 2003; Murray et al., 2012). INs are abundant across freeze-tolerant organisms, such as bacteria, fungi, plants, and lichens, and play fundamental roles in their survival (Maki et al., 1974; Kieft and Lindow, 1988; Pouleur et al., 1992; Lundheim, 2002; Fröhlich-Nowoisky et al., 2015; Eufemio et al., 2023).

The most efficient biological INs described to date are the bacterial plant pathogens, *Pseudomonas syringae*, whose ability to facilitate freezing at exceptionally warm sub-zero temperatures originates from ice nucleating proteins (INPs) located in the cell outer membrane (Govindarajan and Lindow, 1988). *P. syringae* INPs assemble into functional aggregates that are categorized into classes A-C based on freezing temperature

and assembly size (Kozloff et al., 1983; Govindarajan and Lindow, 1988; Turner et al., 1990). Large protein aggregates associated with class A allow the bacteria to achieve IN-activity close to -1°C, while class C consists of comparatively small INPs active at ~ -7.5°C (Lukas et al., 2020; Lukas et al., 2022; Hartmann et al., 2022; Bieber and Borduas-Dedekind , 2024; Renzer et al., 2024).

While there is considerable interest in bacterial INPs, relatively little attention has been paid to lichens, despite several species having been identified as powerful INs that enable ice formation as high as -1.9°C (Kieft and Lindow, 1988; Ashworth and Kieft, 1992; Moffett et al., 2015; Eufemio et al., 2023). Lichen INs are sensitive to protein-degrading treatments, such as high temperatures and urea, implying that, like bacteria, they consist of proteinaceous compounds (Kieft and Ruscetti, 1990). In contrast to most bacterial INPs, lichen INs are cell-free and induce freezing without the need to be anchored in a cell membrane (Kieft and Ruscetti, 1990; Moffett et al., 2015). Therefore, lichens INs are more similar to what has been observed for several fungi (Fröhlich-Nowoisky et al., 2015; Pummer et al., 2015; Kunert et al., 2019), the Gram-positive bacterium *Lysinibacillus parviboronicapiens* (Failor et al., 2017) and bacteria in the Gram-negative genus *Pantoea*, formerly classified as *Erwinia* (Phelps et al., 1986).

Lichens, including their fungal and prokaryotic microbiomes (Arnold et al., 2009; Hodkinson et al., 2012), can tolerate extreme conditions and survive in environments where most organisms cannot. They dominate in nearly 10% of the earth's terrestrial ecosystems (Honegger, 2007; Papazi et al., 2015). Lichen-derived INs, specifically in the form of airborne asexual reproductive propagules (Marshall, 1996; Tormo et al., 2001), have been detected in the atmosphere, where they can contribute to cloud glaciation and trigger precipitation (Henderson-Begg et al., 2009: Moffett et al., 2015). Lichens are mutualistic associations between a fungal symbiont (mycobiont) and one or two photoautotrophic symbionts (photobiont) (Lutzoni and Miadlikowska, 2009). Approximately 85% of lichen-forming fungal species are in association with a green algal photobiont, forming bi-membered lichen thalli. In about 10% of lichen species, the photobiont is exclusively cyanobacterial, forming a different bi-membered association. Only 3-4% of lichenized fungal species are associated with both photobiont partners, forming tri-membered lichen thalli (Honegger, 2007; Nash, 2008; Henskens et al., 2012). Early studies on axenic lichen cultures have identified the mycobiont as IN-active, while the photobiont is comparatively inactive (Kieft and Ahmadjian, 1989).

*Peltigera* lichens are strong INs, with several species initiating freezing at or above -5°C (Eufemio et al., 2023). *Peltigera*, which consists of both bi-membered and tri-membered species, is one of the most widespread lichen genera and is particularly abundant in boreal biomes (Pojar and MacKinnon, 1994; Miadlikowska and Lutzoni, 2000; Martinez et al. 2003; Magain et al., 2017; Magain et al., 2023), making their INs highly relevant to biological and atmospheric processes (Creamean et al., 2021; Moffett et al., 2015). Despite the ecological importance and formidable IN-activity of *Peltigera*, the abundance and efficiency of INs across the genus remain unknown.

Here, we surveyed *Peltigera* species from across a range of biomes spanning the Arctic, the Northwestern United States, and Central and South America for IN activity. We used a high-throughput twin-plate ice nucleation assay (TINA) to quantify the INs of select thalli and to assess the potency of an IN-active culture, L01-tf-B03, isolated from the *Peltigera britannica* JNU22 thallus. We further evaluated the stability of the *Peltigera* INs under freeze-thaw cycles to gain insights into their ability to remain IN-active under environmentally relevant conditions.

## 2 Materials and methods

### 2.1 Sampling

Lichen thalli of the genus *Peltigera* were collected based on availability from prior sampling campaigns in the United States, Canada, Brazil, and Costa Rica (Fig. 1c) between February 2003 and August 2024 (Table 1). *Peltigera* lichens form leaf-like (foliose), typically large and prominent thalli that are relatively easy to identify at the genus level. Their distinct features include the absence of a lower fungal protective layer (cortex) and the presence of a dense cobweb-like fungal layer that forms a network of veins with numerous rhizines. The genus *Peltigera* exhibits two types of symbiotic relationships: a two-partner (bi-membered) association with a cyanobacterium *Nostoc*, and a three-partner (tri-membered) association. In the latter, the green alga *Coccomyxa* serves as the primary photobiont while the cyanobacterium *Nostoc* is contained within specialized structures (cephalodia) on the thallus (Miadlikowska and Lutzoni 2000). Species were identified using a vegetation identification guide (Pojar and MacKinnon, 1994) and recent systematics revisions of the genus (Magain et al. 2017; Magain et al. 2023). Eleven bi-membered (Fig. 1a) and six tri-membered (Fig. 1b) lichens were sampled, representing eight *Peltigera* species (Table 1). Four specimens were collected from rock and tree substrates in temperate rainforests in Juneau, Alaska, USA and Washington, USA, in 2022. Six samples were obtained from the Arctic: Two from Utqiagvik, Alaska, USA, in 2024 and four from Nunavut, Canada, in 2023. Two were collected from boreal forests of Québec, Canada, in 2011. Three lichens were collected in the Atlantic rainforest ecosystem in Minas Gerais, Brazil, in 2012, and two samples were obtained from the Talamanca Mountain Range of Costa Rica in 2003. These locations encompassed a diverse range of biomes across distinct geographic areas (Fig. 1c). The elevations of the collection sites varied from sea level (e.g., in Alaska) to 3400 meters (e.g., in the Talamanca Mountain Range, Costa Rica). Lichen samples were collected based on their availability and accessibility in nature. Collected specimens were either stored in sterile containers at -18°C or kept at room temperature in a dry state in paper bags.

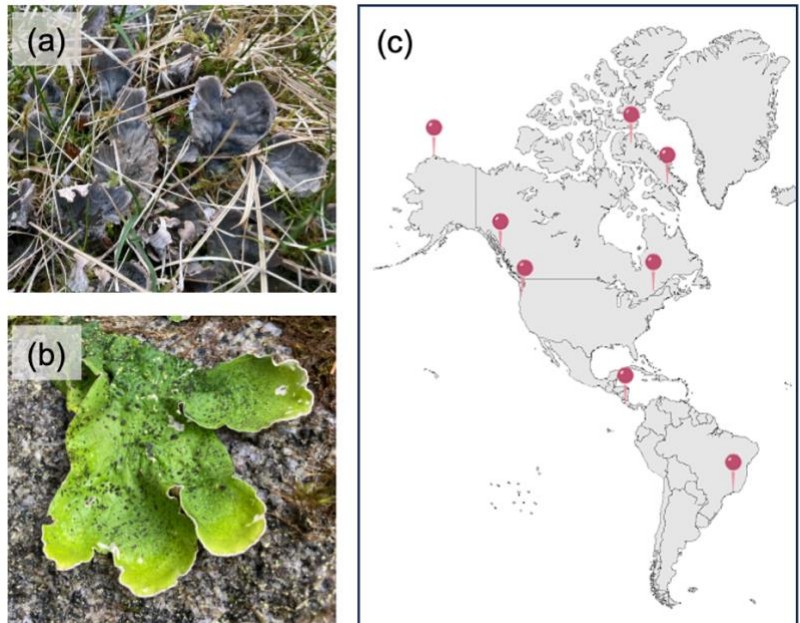

**Figure 1**. Representative images of *Peltigera* lichens showing (a) bi-membered and (b) tri-membered thalli. When wet, tri-membered *Peltigera* species display a distinctive bright green color due to their primary green photobiont *Coccomyxa*, contrary to bi-membered *Peltigera* species which contain a cyanobacterium photobiont (*Nostoc*). (c) Map showing the study's sampling locations.

## 2.2 Isolation of IN-active fungal cultures from *P. britannica* JNU22 thalli

Malt/yeast extract medium (MY medium) with pH 5 (Pichler et al., 2021) was prepared by mixing 22 g L⁻¹ malt yeast extract (Difco™ Malt Extract Broth, Fisher Scientific, Massachusetts, United States) and 20 g L⁻¹ agar (Merck KGaA, Darmstadt, Germany) in water. The medium was sterilized by autoclaving at 121°C for 20 minutes. *P. britannica* JNU22 cultures were cultivated following a procedure adapted from Yoshimura et al. (2002). The lichen thallus was brought to room temperature from -18°C and washed under pure water for 10 minutes to remove as many epiphytic micro-organisms as possible. Two methods were employed to isolate the lichen-associated organisms. In the first method, pieces of thallus approximately 1 mm² were cut out using a sterilized scalpel blade and placed on MY medium in Petri dishes labeled L01-tf-A and L01-tf-B. The plates were referred to as the 'thallus fragments' (tf). In the second method, three 1 mm² thallus pieces were ground to a pulp in 300 μL pure water. This was referred to as a 'thallus slurry' (ts). The slurry was spread on the surface of MY medium in plates labeled L01-ts-A and L01-ts-B and grown for 14 days at room temperature. Both methods produced sufficient growth to allow for subculturing onto new MY medium plates. Sterile tweezers were used to transfer growth from the center of each plate onto clean MY plates and the sub-culturing process was repeated until fourteen pure cultures were obtained. All procedures were carried out under sterile conditions. The pure cultures were grown at room temperature for up to 14 days,

then kept at 4°C for up to 8 weeks. An optical microscope (Zeiss Axioscope 5, Carl Zeiss Microscopy GmbH, Jena, Germany) was used to visually identify cultures as mycelial fungi, yeasts, and molds (select images shown in Supplemental Fig. 4). Subsequent measurements of ice nucleation activity were conducted to identify IN-active cultures (described in Sect. 2.3 and 2.4).

145

## 2.3 Purification of aqueous IN extracts

The *Peltigera* thalli that were stored in a frozen state were prepared for quantitative analysis by washing the thallus in pure water (Millipore Milli-Q® Simplicity 185 Water Purification system, Merck KGaA, Darmstadt, Germany) to minimize contamination from external sources of ice nucleating particles. To ensure consistency in the water content of the thalli, specimens that were stored in a dehydrated, dormant state were weighed, rehydrated in pure water for 30 minutes, and reweighed prior to washing. Extracts were prepared for ice nucleation assays using a standardized procedure previously described by Eufemio et al. (2023). In short, 2 g of lichen thallus in 5 mL of pure water were ground to a fine pulp. The pulp was centrifuged at 5000 rpm for 10 min and the supernatants were filtered through 0.22 μm pore diameter syringe filters (Millex® Syringe Filter, Merck KGaA, Darmstadt, Germany). The resulting aqueous extracts contained molecules that were both secreted and bound to the cell wall.

Aqueous extracts obtained from pure cultures isolated from *P. britannica* JNU22 were prepared as described by Kunert et al. (2019) with the following modifications. The cultures were collected from the center of each plate and placed into a sterile Eppendorf tube, and the weight of the collected material was determined. A primary suspension was made by suspending 10 mg of harvested material in 1 mL of pure water. The suspensions were vortexed three times at 2700 rpm for 1 min, then filtered through 0.22 μm pore diameter syringe filters. The resulting extracts contained ice nucleators from those cultures.

## 2.4 Ice nucleation experiments

The aqueous extracts from *Peltigera* thalli and the *P. britannica* JNU22 cultures were tested for ice nucleation activity immediately after purification using a Vali-type set-up (Vali, 1971). 20 droplets (1 μL) of extract were cooled at 3°C min$^{-1}$ from 0°C to -20°C on a temperature-controlled aluminum plate (Linkam Scientific Instruments LTD, United Kingdom). The freezing temperature of each droplet was identified based on the optical change in appearance that occurred with freezing. The temperature at which 50% of the droplets froze, $T_{50}$, was manually recorded. The IN-activity of the positive control, 1 mg mL$^{-1}$ of inactivated *P. syringae* ($T_{50}$ of -3.5°C) and water (the negative control with a $T_{50}$ of -11°C) was measured using experimental parameters

identical to those of the lichen samples. Aliquots of MY medium were used as negative controls for the ice nucleation assays of the cultures obtained from the *P. britannica* JNU22 thallus, and froze at the same $T_{50}$ value (-11°C) as the background water. We cannot definitively attribute the freezing to either the MY medium or the water. Triplicate droplet freezing experiments were conducted.

Although the Vali-type apparatus was sufficient for crude initial tests of ice nucleation activity, more statistically rigorous measurements were needed for quantitative analysis of the extracted INs. High-throughput ice nucleation experiments were performed using TINA, as described by Kunert et al. (2018). 1 mL of each investigated IN extract was serially diluted in 10-fold increments with a liquid-handling station (epMotion ep5073, Eppendorf, Hamburg, Germany). 96 droplets (3 µL) per dilution were placed in two 384-well plates, which were cooled at a continuous rate of 1°C min$^{-1}$ from 0°C to -30°C. In the TINA set-up, the negative control of pure water had a $T_{50} \sim$ -23.5°C. For each experiment, the droplet-freezing temperatures were extracted, and the fraction of frozen droplets at different temperatures was used to calculate the cumulative number of active INs per unit mass of sample ($N_m$) using Vali's equation (Vali, 1971).

## 2.5 Treatments of aqueous extracts

Freeze-thaw cycles were used as a measure of IN stability against temperature fluctuations. Aliquots of aqueous lichen extract were frozen by cooling to -30°C at a rate of 1°C min$^{-1}$ and thawed to room temperature up to 6 times over the course of 24 hours. The IN-activity was measured using TINA after each cycle.

## 3 Results

### 3.1 IN-activity is widespread in *Peltigera* lichens

While several lichen species were previously identified as powerful INs (as shown in Supplemental Fig. S1) (Kieft and Lindow, 1988; Eufemio et al., 2023), the prevalence of IN-activity within the genus *Peltigera* remained largely unknown. Two assays, a Vali-type droplet freezing assay and TINA, were performed to better evaluate the frequency of IN-activity across *Peltigera* species. Table 1 presents the freezing temperatures of 17 lichen extracts measured by the Vali-type droplet freezing assay, 9 of which were confirmed by TINA. The initial solutions had a concentration of 0.4 g mL$^{-1}$ and were then serially diluted 10-fold. We find that all *Peltigera* extracts freeze between -4.4°C and -11.2°C. The type of symbiosis, i.e., whether the lichens are tri-membered or bi-membered, does not appear to have an influence on the ice nucleation. Notably, the two most IN-active lichens, the tri-membered *P. britannica* JNU22 and bi-membered

*P. austroamericana* 34529, show only a 0.7°C difference in IN-activity. *P. britannica* JNU22 has a TINA $T_{50}$ value of -5.1°C while *P. austroamericana* 34529 freezes at -4.4°C. Additionally, the IN-activity of lichens does not seem to strongly correlate with the geographic region or climate zone where the samples were collected. For example, TINA measurements reveal that the ice nucleation temperature of *P. aphthosa* PL729, which was obtained from the Canadian Arctic, and *P. neopolydactyla* JNU22, which was collected in the temperate rainforests of the Northwestern United States, varied by only 0.6°C despite significant ecosystem differences. A larger sample size is needed to conclusively determine whether climatic or geographic trends in IN-activity exist. Given the range of environments in which IN-active *Peltigera* lichens grow, it is possible that the ice nucleation ability is an incidental property (Lundheim, 2002; Moffett et al., 2015). However, considering the high IN-activity and frost tolerance of lichens, we cannot dismiss that the INs provide a specific physiological benefit. Because the specimens were collected between 2003 and 2024 and subsequently stored in either a frozen or dehydrated state, we cannot exclude that the age of the sample or the method of storage impacted the ice nucleation. However, Table 1 provides strong evidence that all tested *Peltigera* lichens contain active INs, and we conclude that ice nucleation activity is a common trait across the genus *Peltigera*.

We observe that the freezing spectra of selected *Peltigera* (Fig. 2) indicate the presence of two distinct activation temperatures, which agrees with prior measurements of lichen INs (Eufemio et al., 2023). *P. britannica* JNU22 consistently has an initial activation temperature of ~ -5°C and a second of ~ -12.5°C, with a plateau between ~ -7 and -12.5°C. Similarly, both *P. neckeri* PNW22 and *P. aphthosa* PL729 INs reveal two activation temperatures at ~ -6°C and ~ -11°C. *P. austroamericana* 34529 INs initially induce freezing at ~ -4°C and a slight rise in the freezing spectra at ~ -7°C indicates a second population of active INs. The INs responsible for the two rises in the freezing spectra have been previously assigned as class 1, which contributes to the initial freezing, and class 2, which refers to INs active at the lower temperature (Eufemio et al., 2023). The differential freezing spectra (de Almeida Ribeiro et al., 2023) of *P. britannica* JNU22 further confirms the presence of both classes (shown in Supplemental Fig. S2). All *Peltigera* INs were found to be susceptible to heat treatments (Supplemental Fig. S3), supporting previous findings that lichen INs are proteinaceous, at least in part (Kieft and Ruscetti, 1990). Some species appear to be highly sensitive to heat, while others are less affected. For example, *P. aphthosa* PL729 activity is substantially lowered from -6.7°C to -9.3°C while *P. membranacea* PNW22 efficiency decreases by only 1.1°C. The differences in activation temperatures and heat sensitivity across *Peltigera* species may indicate variations in the macromolecular composition of INs. However, further experiments, including (bio)chemical analyses, are needed to decipher the molecular nature of the INs and to identify whether classes 1 and 2 are due to an aggregation mechanism similar to the bacterial INP classes A and C.

**Table 1.** Ice nucleation activity of undiluted *Peltigera* lichen extracts determined using a Vali-type (initial) and high-throughput ice nucleation assay (TINA). Freezing temperatures, $T_{50}$, are defined as the temperature at which 50% of the extract droplets are frozen. Samples with sufficient volume for a full dilution series were measured with TINA. Extracts labeled N/A were not measured using TINA. Species are arranged according to symbiosis type (tri- or bi-membered) and collection location.

| *Peltigera* Species | Symbiosis Type | Collection Location | Collection Date | Initial $T_{50}$ (°C) | TINA $T_{50}$ (°C) |
|---|---|---|---|---|---|
| *P. britannica* JNU22 | Tri-membered | Alaska, USA | 2022 | -5.7 | -5.1 |
| *P. aphthosa* BRW1 | Tri-membered | Alaska, USA | 2024 | -6.8 | N/A |
| *P. aphthosa* BRW2 | Tri-membered | Alaska, USA | 2024 | -6.9 | N/A |
| *P. aphthosa* PL729 | Tri-membered | Nunavut, Canada | 2023 | -6.7 | -6.7 |
| *P. aphthosa* PL708 | Tri-membered | Nunavut, Canada | 2023 | -6.8 | N/A |
| *P. aphthosa* P5057 | Tri-membered | Quebec, Canada | 2011 | -8.3 | -11.2 |
| *P. neopolydactyla* JNU22 | Bi-membered | Alaska, USA | 2022 | -6.3 | -6.1 |
| *P. membranacea* PNW22 | Bi-membered | Washington, USA | 2022 | -7.3 | -5.8 |
| *P. neckeri* PNW22 | Bi-membered | Washington, USA | 2022 | -7.5 | -5.9 |
| *P. malacea* PL744 | Bi-membered | Nunavut, Canada | 2023 | -6.8 | -6.2 |
| *P. neckeri* PL713 | Bi-membered | Nunavut, Canada | 2023 | -6.6 | -7.1 |
| *P. neopolydactyla* P0309 | Bi-membered | Quebec, Canada | 2011 | -6.9 | N/A |
| *P. austroamericana* 34390 | Bi-membered | Brazil | 2012 | -9.3 | N/A |
| *P. austroamericana* 34529 | Bi-membered | Brazil | 2012 | -4.9 | -4.4 |
| *P. dolichorrhiza* 34433 | Bi-membered | Brazil | 2012 | -8.1 | N/A |
| *P. dolichorrhiza* CR-8 | Bi-membered | Costa Rica | 2003 | -8.8 | N/A |
| *P. dolichorrhiza* CR-4 | Bi-membered | Costa Rica | 2003 | -8.7 | N/A |

### 3.2 *Peltigera* INs retain activity with freeze-thaw cycles

Several lichen species outside of the genus *Peltigera* have been found to maintain freezing efficiency after exposure to conditions associated with high altitudes, such as rapidly changing temperatures (Eufemio et al., 2023). Figure 2 displays the effects of consecutive freeze-thaw cycles on aqueous extracts of *P. britannica*

JNU22, *P. neckeri* PNW22, *P. aphthosa* PL729, and *P. austroamericana* 34529, which were collected in Alaska and Washington, USA, northern Canada, and Brazil, respectively. The freezing spectra show that the lichens retain ice nucleation across six cycles, independent of the geographic region of origin. Across all cycles, the cumulative number of INs remains nearly constant. The negligible impact of consecutive freeze-thaw cycles on the ice nucleation activity highlights the stability of *Peltigera* INs and emphasizes both their likelihood to survive the aerosolization process and their capacity to act as ice nucleating agents in the atmosphere.

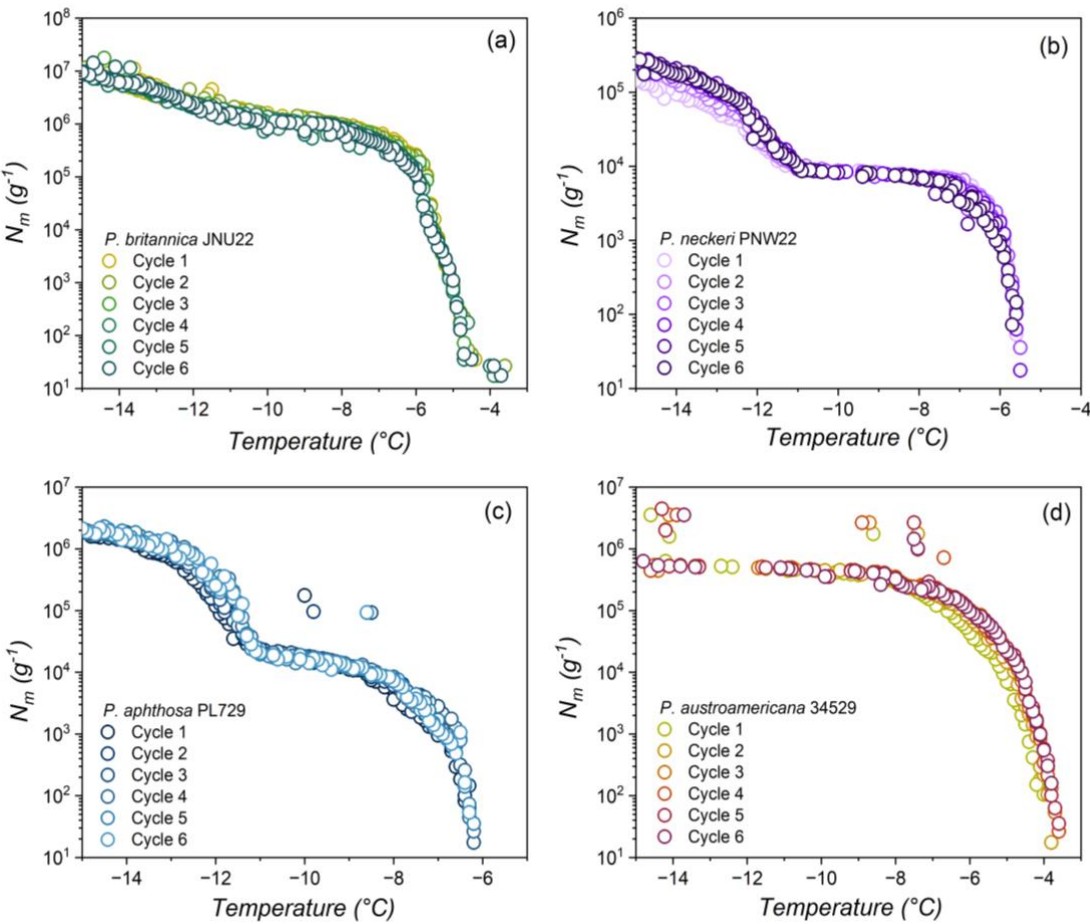

**Figure 2**. Effects of freeze-thaw cycles on *Peltigera* lichen ice nucleation activity. Shown are the cumulative number of INs per unit mass ($N_m$) of (a) *P. britannica* JNU22, (b) *P. neckeri* PNW22, (c) *P. aphthosa* PL729 and (d) *P. austroamericana* 34529 across 6 consecutive cycles.

### 3.3 Isolation of IN-active lichen-associated organisms

Besides the primary mycobiont and photobiont partners, lichens are host to a diverse community of additional micro-organisms, including fungi and bacteria (Arnold et al., 2009; Hodkinson et al., 2012). We surveyed *P. britannica* JNU22-associated bionts for ice nucleation to investigate the relative contributions of each lichen-

associated organism to the lichen freezing activity. Fourteen pure cultures were isolated from the thallus of *P. britannica* JNU22 and identified as fungi, including molds and yeasts, based on the morphology and microscopic images (see Supplemental Fig. S4 for select images). Table S1 presents the freezing temperatures of the aqueous extracts containing INs of each culture as measured by the Vali-type droplet freezing assay. Initial screenings reveal substantial variations in the IN-activity of the cultures. The aqueous extract containing INs from the most active culture, labeled L01-tf-B03, induced freezing at -5.2°C. Based on the presence of mycelial-like growth, we classified L01-tf-B03 as a lichen-associated fungus. It is notoriously difficult to isolate lichen mycobionts in pure culture (Cornejo et al., 2015). Moreover, they are very slow growing compared to most fungi. As far as we know, *Peltigera* has never grown successfully in culture. However, *Peltigera* lichen thalli are well-known to host diverse communities of endolichenic fungi, which are frequently isolated in pure culture from *Peltigera* thalli (Arnold et al., 2009; U'Ren et al., 2010; U'Ren et al., 2012). Compared to lichen mycobionts, most endolichenic fungi grow quickly. Endolichenic fungi are also referred to as lichen-associated fungi, which encompass all fungi associated with lichen thalli other than the lichen mycobiont. The least active culture, L01-tf-B01, appeared to be a yeast. The aqueous IN extract of L01-tf-B01 did not freeze until -9.6°C.

### 3.4 The L01-tf-B03 culture is the most potent ice nucleator

Figure 3a displays the results of TINA measurements of an aqueous extract of INs from the mycelial surfaces of L01-tf-B03. The initial concentration was 1 mg mL$^{-1}$ and a complete dilution series was performed. The freezing spectra of L01-tf-B03 confirm the potency of the culture as determined by the Vali-type assay (Table S1 in the Supplement). We observe two rises in the spectra of L01-tf-B03 at ~ -5.6°C and ~ -6.5°C.

Figure 3b shows that at 1 mg mL$^{-1}$, the T$_{50}$ value of L01-tf-B03 is approximately -5.6°C. Impressively, the extract retains strong IN-activity as low as 1 μg mL$^{-1}$, with the T$_{50}$ decreasing to only ~ 5.8°C. A minor decrease in IN-activity occurs at a concentration of 0.1 μg mL$^{-1}$, at which the T$_{50}$ lowers from -5.8°C to ~ -6.3°C. However, it is not until the extract is diluted to below 0.1 ng mL$^{-1}$ that significant impacts on the IN-activity are observed. 0.1 ng mL$^{-1}$ corresponds to a T$_{50}$ of ~ -7.2°C, after which the T$_{50}$ values are dramatically lowered and are similar to pure water (TINA T$_{50}$ of ~ -23.5°C). The potency of L01-tf-B03 is evident in comparison to live bacteria from the strain *P. syringae* Cit7 (Renzer et al., 2024), which was subjected to the same dilution series as the lichen culture. We define potency as the concentration of INs necessary to observe IN-activity at temperatures above the freezing point of background water (~ -23.5°C). Efficiency refers to the highest temperature at which an IN induces freezing, regardless of its concentration. While *P. syringae* INs are highly efficient at 1 mg mL$^{-1}$, with a T$_{50}$ of ~ -2°C, the IN-activity is rapidly lost with subsequent

dilutions. At a concentration of 0.01 mg mL$^{-1}$, the T$_{50}$ is lowered to ~ -3.1°C, and at 0.1 µg mL$^{-1}$ the IN-activity decreases further to -7.7°C. The large decrease of over 4°C in bacterial freezing efficiency is in striking contrast to L01-tf-B03, for which the IN-activity is reduced by less than 1°C at the same concentration. It appears that L01-tf-B03 maintains class 1 INs at this concentration, while *P. syringae* INs have already shifted to class C. We speculate that L01-tf-B03 possesses a mechanism to stabilize aggregates which enable high IN-activity. At 1 ng mL$^{-1}$, the IN-activity of *P. syringae* is eliminated and the T$_{50}$ values are comparable to pure water. At the lowest measured dilution, L01-tf-B03 contains a notably larger cumulative number of active INs than *P. syringae* (details can be found in Fig. S5 in the Supplement).

Our findings suggest that the L01-tf-B03 INs are 1000 times more potent than those of *P. syringae,* given that the total decay of *P. syringae* IN-activity occurs at a concentration of 1 ng mL$^{-1}$ while L01-tf-B03 activity is not fully eliminated until 0.01 ng mL$^{-1}$. It is worth noting that the IN-activity of *P. syringae* was measured based on the mass of the bacterial cells, whereas for L01-tf-B03, the measurements were based on the mass of the INs released from the mycelium. To the best of our knowledge, the ability of L01-tf-B03 to retain IN-activity at exceptionally low concentrations makes it the most potent documented IN to date. We aim to confirm the genetic identity of L01-tf-B03 in future studies.

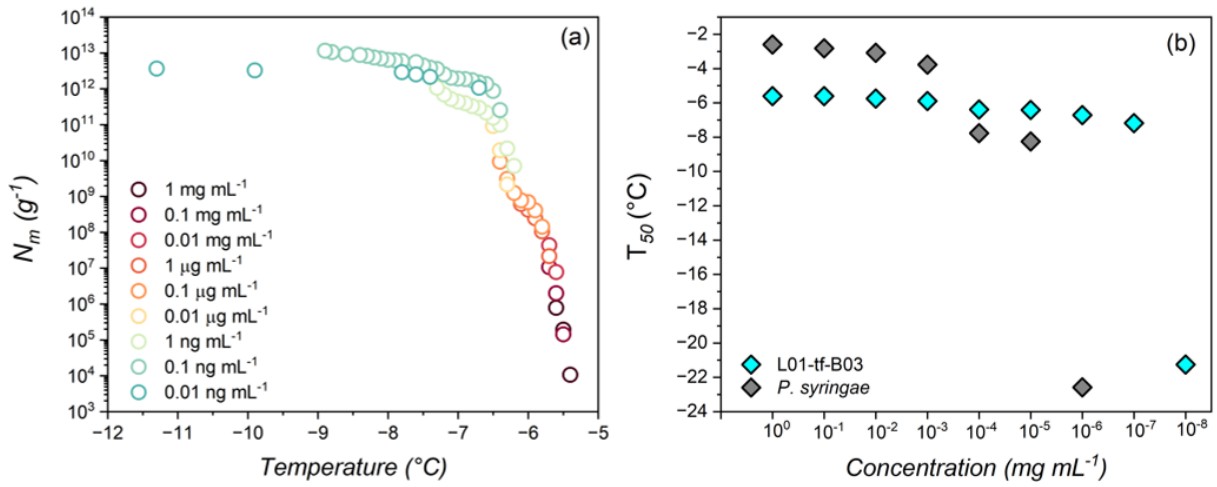

**Figure 3.** (a) Freezing experiment of aqueous extract containing INs (1 mg mL$^{-1}$ to 0.01 ng mL$^{-1}$) from L01-tf-B03. Shown are the cumulative number of INs per unit mass ($N_m$) of L01-tf-B03. (b) Dilution effects on the IN-activity of L01-tf-B03 (cyan) aqueous extract and *P. syringae* in water (gray). The concentration range is 1 mg mL$^{-1}$ to 0.01 ng mL$^{-1}$. *P. syringae* INs are inactive at concentrations below 1 ng mL$^{-1}$ and are not shown.

**4 Conclusions**

Ice nucleation activity of *Peltigera* lichens was investigated in a Pan-American survey across tropical to arctic environments. All the lichens were IN-active above -12°C and 35% initiated freezing at or above -6.2°C. This threshold highlights the remarkable ice-nucleating capability of *Peltigera* lichens, compared to other known fungal ice nucleators (Pouleur et al., 1992; Fröhlich-Nowoisky et al., 2015; Kunert et al., 2019; Schwidetzky et al., 2023). We find no relationship between the symbiosis type and freezing activity. Further, there is no apparent correlation between biogeographic patterns and lichen IN-activity. Our findings illustrate that potent ice nucleation may be a common trait across *Peltigera*. A concentration series of select samples exposed to repetitive freeze-thaw cycles revealed that *Peltigera* INs maintain activity under experimental conditions that serve as a proxy for high altitudes. Given the presence of *Peltigera* in many world regions, the lichen INs may be prevalent in airborne fungal communities where they could remain active and exert a sustained influence on atmospheric processes (Henderson-Begg et al., 2009: Moffett et al., 2015: Womack et al., 2015).

The *P. britannica* JNU22 isolate, L01-tf-B03, was identified as a lichen-associated fungus. These findings are consistent with previous measurements of mycobiont IN-activity by Kieft and Ahmadjian (1989), in which the fungal cultures were found to produce more active INs than the photobionts. L01-tf-B03 induced warm sub-zero ice formation at -5.6°C at 1 mg mL$^{-1}$ and retained IN-activity of ~ -7.2°C at concentrations as low as 0.1 ng mL$^{-1}$. Our results show that the INs released from the fungal culture are nearly 1000 times more potent than the most IN-active bacterial cell-anchored INs classified to date. While the slow growth and limited commercial availability of lichens makes them impractical as a replacement for *P. syringae* in products like Snomax (inactivated *P. syringae*; Snomax Int.), the remarkable IN-activity of the fungal strain L01-tf-B03 at exceptionally low concentrations suggests it could be worth exploring its potential for use in commercial applications. Future research should investigate further methods to sustainably cultivate or harness these ice-nucleating components for industrial or technological purposes.

*Code/Data availability*. All data are available from the corresponding author upon request.

*Author contribution*. KM and RJE designed the experiments. TLS, JM, and FL provided lichen samples and identified them. RJE and GR performed the experiments. MR, BAV, JM, and FL provided guidance on culture growth and lichen ecology. RJE, KM, GR, BAV, and FL discussed the results. RJE and KM wrote the paper with contributions from all co-authors.

*Competing interests*. The authors declare that they have no conflict of interest.

*Acknowledgements.* We are grateful to the MaxWater Initiative from the Max Planck Society. K. M. acknowledges support by the NSF under grant no. NSF (2308172, 2116528) and from the Institutional Development Awards (IDeA) from the National Institute of General Medical Sciences of the NIH under grants # P20GM103408 and #P20GM109095. Biorender was used for image creation (Fig. 1c). We thank Mischa Bonn for stimulating discussions and help with manuscript revisions. Field sampling of *Peltigera* specimens was possible through NSF funding (0133891, 1025930, 1046065, and 2031927) to FL and JM. Sampling in Iqaluit and Pond Inlet (Nunavut, Canada) was possible through the collaboration and permission from the Nunavut Research Institute, and Nunavut Department of Environment, Wildlife Research Permit 2023-057. We are very grateful to the Hunters and Trappers Organizations in Iqaluit and Pond Inlet for approving our research plan in their communities during the summer of 2023.

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
