# Peer review of "Peltigera lichen thalli produce highly potent ice nucleating agents"

_EGUsphere, 2024_

## Author Comment (AC1)

**Response to Referee #1**

This article presents investigations into lichen of the genus Peltigera as producers of ice nucleators (INs). It is well-conceived, methodologically sound, and enjoyable to read. The insights gained through these investigations are new and interesting, so they merit publication in Biogeosciences. There are a few minor issues I recommend the authors to consider in a revision:

**Response**: We thank the reviewer for carefully reading our manuscript. Below we addressed all comments, point-by-point.

1) Peltigera are mostly ground dwelling and have a compact morphology. Which process could dislocate particles small enough from them to escape the surface layer and reach higher altitudes? I would appreciate to a sentence or two on that issue in in the Conclusions.

**Response**: The reviewer makes a valid point that *Peltigera* are nearly exclusively ground-dwelling, but many lichens are epiphytic, or grow on rocks, and are often dominant in tundra and alpine habitats (Pojar and MacKinnon, 1994; Nash, 2008). Lichens are also common in cloud forests, including cyanolichens. *Peltigera* are found in all these habitats. In addition, asexual reproductive propagules of lichen are airborne. Therefore, lichen propagules are a likely source of lichen-derived INs at higher altitudes.

**Action taken**: We added additional text and references to the introduction and conclusions of the manuscript. Starting on line 68 of the Introduction, the text now reads, "Lichen-derived INs, specifically airborne asexual reproductive propagules (Marshall, 1996; Tormo et al., 2001), have been detected in the atmosphere, where they can contribute to cloud glaciation and trigger precipitation (Henderson-Begg et al., 2009: Moffett et al., 2015)". Line 323 in the Conclusions now reads, "Given the presence of *Peltigera* in many world regions, the lichen INs may be prevalent in airborne fungal communities where they could remain active and exert a sustained influence on atmospheric processes (Henderson-Begg et al., 2009: Moffett et al., 2015: Womack et al., 2015)".

2) Lines 31-33 and lines 330-333 state: "Our analysis suggests that the INs released from this fungus in culture are 1000 times more efficient than the most potent bacterial INs from Pseudomonas syringae." I find the term "efficient" problematic in this context because "efficient"

often refers to the activation temperature of INs, i.e., INs active > -10°C (e.g., Zhang et al., 2020, https://doi.org/10.1016/j.atmosres.2020.105129). You also define T50 as a measure of efficiency (lines 168-169).

**Response**: We thank the reviewer for these important points. We have updated our definitions and now define potency as the concentration required to induce ice nucleation at temperatures above the background freezing point (e.g., ~-23.5°C). We further agree that our current definition of $T_{50}$ does not align with our use of the term "efficiency" in this manuscript. The $T_{50}$ value is the temperature at which 50% of droplet freeze in the droplet-freezing assay and a standard measure of ice-nucleation activity, rather than a measure of efficiency.

Efficiency is now defined throughout the manuscript as the highest temperature at which an ice nucleator induces freezing, regardless of its concentration. By this definition, *P. syringae* is more efficient since it has the highest IN-activity, but L01-tf-B03 is 1000 times more potent, given that the total decay of *P. syringae* IN-activity occurs at a concentration of 1 ng mL$^{-1}$ while L01-tf-B03 activity is not fully eliminated until 0.01 ng mL$^{-1}$.

**Action taken**: Based on the recommendation of the reviewer, we have made several changes within the manuscript and also altered the title of the paper. Specifically, we changed 'efficiency' to 'potency' on line 280, which now reads, "The potency of L01-tf-B03 is evident in comparison to the live bacteria from the strain *P. syringae* Cit7 (Renzer et al., 2024), which was subjected to the same dilution series as the lichen culture".

Specific definitions of potency and efficiency have been added on line 283, which now reads, "We define potency as the concentration of INs necessary to observe IN-activity at temperatures above the freezing point of background water (~ -23.5°C). Efficiency refers to the highest temperature at which an IN induces freezing, regardless of its concentration". In Section 3.4, the term 'efficiency' has been changed to 'potency' in every instance used to describe the IN-activity of L01-tf-B03. We also updated the wording in the Abstract and Conclusions. Line 32 now reads, "Our analysis suggests that the INs released from this fungus in culture are 1000 times more potent than the most IN-active bacterial INs from *P. syringae*". Line 331 has been updated to, "Our results show that the INs released from the fungal culture are nearly 1000 times more potent than the most IN-active bacterial cell-anchored INs classified to date".

We additionally edited the text on line 169 to read, "The freezing temperature of each droplet was identified based on the optical change in appearance that occurred with freezing and the temperature at which 50% of the droplets froze, $T_{50}$, was recorded."

3) Line 113: What is meant with "bioavailability" here?

**Response**: We acknowledge that the use of 'bioavailability' may be unclear in this context. Therefore, we have changed the wording to clarify that samples were collected based on the availability of these lichens in nature.

**Action taken**: We updated the text on line 113 to read, "Lichen samples were collected based on their availability and accessibility in nature."

4) Lines 168-169: "The temperature at which 50% of the droplets froze, T50, was recorded as a measure of the efficiency of the INs." I guess it means the same as "freezing efficiency", an expression first used in line 201? If so, please add (in brackets) this term to the end of the sentence in lines 168-169.

**Response**: We thank the reviewer for addressing the inconsistency in our definitions. We replaced the term "freezing efficiency" with "IN-activity" to improve clarity. As addressed above in point #2, we also updated the definition of $T_{50}$.

**Action taken**: The text on line 201 has been updated and now reads, "Notably, the two most IN-active lichens, the trimembered *P. britannica* JNU22 and bimembered *P. austroamericana* 34529, show only a 0.7°C difference in IN-activity." Additionally, line 169 now reads, "…the temperature at which 50% of the droplets froze, $T_{50}$, was recorded."

5) Line 170: The T50 value of water is relatively high (-11°C). Were values of Peltigera samples corrected for that and, if so, how?

**Response**: We thank the reviewer for raising this point. We recognize that this $T_{50}$ value, in the Vali-type ice nucleation assay set-up, is relatively high. However, because all the *Peltigera* samples froze at $T_{50}$ values well above -11°C, the background freezing temperature of water was not corrected for. In addition, inactivated *P. syringae* (Snomax) was consistently measured as an internal control and froze at ~ -3.5°C to ensure that the freezing was due to the presence of an ice

nucleator and not due to the water or environmental conditions in the lab. Most importantly, the Vali-type assay is only used for a prescreening of the crude estimates of IN-activity prior to more statistically rigorous TINA measurements.

**Action taken**: We added text on line 172 specifying that the Vali-type assay is used for crude, initial measurements. This line now reads, "Although the Vali-type apparatus was sufficient for crude initial tests of IN-activity, more statistically rigorous measurements were needed for quantitative analysis of the extracted INs."

6) Line 173: Consider replacing "robust" with "precise".

**Response**: We thank the reviewer for the suggested improvement. The text has been revised to clarify that TINA measurements are more rigorous with higher statistics than the Vali-type assays.

**Action taken**: The line now reads, "Although the Vali-type apparatus was sufficient for crude initial tests of IN-activity, more rigorous measurements with higher statistics were needed for quantitative analysis of the extracted INs."

7) Lines 212-216: The 96 droplets in TINA experiments may be large enough a number to derive differential IN-spectra from (see Vali, 2019, https://doi.org/10.5194/amt-12-1219-2019). Differential spectra afford clearer interpretation than cumulative spectra, especially in the context of your study.

**Response**: We agree that differential spectra offer valuable insights into the underlying number of IN subpopulations. However, because the primary focus of the manuscript is on the frequency and efficiency of *Peltigera* IN-activity, rather than specific characterization of the ice nucleators, we did not include the differential freezing spectra in the main manuscript. Given the additional interpretation of the cumulative spectra that the differential spectra offers, we have revised the text and added a differential freezing spectra of *P. britannica* JNU22 in the Supplement.

**Action taken**: We added text on line 219 that reads, "The differential freezing spectra of *P. britannica* JNU22 further confirms the presence of both classes". The Supplement has been updated to include the differential freezing spectra of *P. britannica* JNU22 as determined by the heterogeneous underlying-based (HUB) stochastic optimization analysis (de Almeida Riberio et al., 2023).

8) Table 1: Isn't it surprising that the warmest T50 was found in a species collected in the tropics? Could this be taken as an indication for IN production in Peltigera being similarly incidental as it seems to be in pollen (Kinney et al., 2024, doi.org/10.5194/egusphere-2023-2705)?

**Response**: We thank the reviewer for providing this insight. While it is possible that ice nucleation in lichens is an incidental property, we believe it is premature to dismiss the idea that the INs serve a specific biological purpose. Lichens are freeze-tolerant organisms, in contrast to many of the plants that produce pollen, and the freeze tolerance suggests that the ability to nucleate ice could provide ecological or physiological benefits to lichens.

**Action taken**: We added text starting on line 206 that reads, "Given the range of environments in which IN-active *Peltigera* lichens grow, it is possible that ice nucleation ability is an incidental property (Lundheim, 2002; Moffett et al., 2015). However, considering the high IN-activity and frost tolerance of lichens, we cannot dismiss that the INs provide a specific physiological benefit."

9) There appears to be a contradiction in lines 263-265: "Based on the fast growth rate and presence of mycelial-like growth, we classified L01-tf-B03 as a lichen-associated fungus. It is notoriously difficult to isolate mycobionts (Cornejo et al., 2015), which are very slow growing, ..." Why should the fast growth rate seen in L01-tf-B03 support your classification when mycobionts are very slow growing?

**Response**: We thank the reviewer for bringing our attention to this point. Lichen-forming fungi grow very slowly compared to most fungi. Because the L01-tf-B03 culture grew more quickly than expected for the *Peltigera* mycobiont, we classified the culture as a lichen-associated fungus, i.e., not *Peltigera*. We have revised the text to clarify this point.

**Action taken**: We have added references and revised the text on line 264 to read, "It is notoriously difficult to isolate lichen mycobionts in pure culture (Cornejo et al., 2015). Moreover, they are very slow growing compared to most fungi. As far as we know, *Peltigera* was never grown successfully in culture. However, *Peltigera* lichen thalli are well-known to host diverse communities of endolichenic fungi, which are frequently isolated in pure culture from *Peltigera* thalli (Arnold et al., 2009; U'Ren et al., 2010, U'Ren et al., 2012). Compared to lichen mycobionts, endolichenic fungi grow quickly. Endolichenic fungi are also referred to as lichen-associated

fungi, which encompass all fungi associated with lichen thalli other than the lichen mycobiont. For all these reasons, we classified L01-tf-B03 as a lichen-associated fungus."

10) Lines 279-280: The T50 value of -23.5°C indicated here is much lower than the one mentioned in line 170 (-11°C). Please clarify.

**Response**: The variation in freezing temperatures in due to the different systems used for freezing assays – in the Vali-type freezing assay, used for initial measures of IN-activity, water froze at a $T_{50}$ of -11°C. In the TINA set-up, which we used for more robust IN-activity measurements, the water control froze at a $T_{50}$ value of -23.5°C.

**Action taken**: We updated the text on line 280 to "TINA $T_{50}$-value ~ -23.5°C" to clarify which freezing assay was used for the measurements shown in Fig. 3.

11) Lines 284-286: " The large decrease of over 4°C in bacterial freezing efficiency is in striking contrast to L01-tf-B03, for which the IN-activity is reduced by less than 1°C at the same concentration." I think this finding merits an attempt at interpretation.

**Response**: Membrane-supported bacterial ice nucleators are grouped in classes A-C, where large aggregates associated with class A enable IN-activity close to 0°C and class C consists of smaller INPs active ~ -7.5°C. Our current understanding suggests that, unlike bacteria, lichen ice nucleators are cell-free secreted molecules, yet, similar to bacteria, lichens also contain two classes (referred to as class 1 and 2) which contribute to the initial and lower freezing events, respectively. We speculate that, like bacteria, the two freezing temperatures are dependent on functional aggregation. L01-tf-B03 appears to maintain class 1 INs even at exceptionally low concentrations, by which point *P. syringae* INPs have shifted to class C. Our findings may suggest that L01-tf-B03 possesses a non-membrane dependent mechanism to stabilize aggregates. However, future experiments are needed to adequately characterize the molecular basis of the two classes.

**Action taken**: We added text on line 286 that reads, "It appears that L01-tf-B03 maintains class 1 INs at this concentration, while *P. syringae* INs have already shifted to class C. We tentatively speculate that L01-tf-B03 possesses a mechanism to stabilize the highly IN-active aggregates".

---

## Author Comment (AC2)

**Response to Referee #2**

The authors extend their study from Eufemio et al. 2023 which also investigated biological entities as sources of ice nucleators. Their first paper reported a range of lichen species across Alaska, while this manuscript focuses on *Peltigera* lichens from various regions, highlighting their efficiency as IN agents. The new knowledge is therefore incremental, but the authors can improve their story line by addressing the following criticisms and specific comments.

**Response**: We thank the review for carefully reading our manuscript and for the important suggestions, which we addressed below. However, we also respectfully disagree with the statement that the knowledge of this manuscript is incremental.

The Eufemio et al., 2023 paper served as a broad survey of the ice nucleation activity of multiple different lichen species across Alaska. The scope of the current manuscript has been narrowed specifically to investigate ice nucleation in *Peltigera* lichens, a genus known for its global abundance and ecological relevance but largely unstudied in the context of ice nucleation.

Moreover, our discovery that *Peltigera britannica* thalli produce the most potent ice nucleator ever documented (in terms of its ability to retain IN-activity at extremely low concentrations) is exceptional and leads us to strongly disagree with the reviewer's point that the new knowledge is incremental.

Major criticism to address prior to publication decision:

1) The conclusion that "Our analysis suggests that the INs released from this fungus in culture are 1000 times more efficient than the most potent bacterial INs from Pseudomonas syringae" (in abstract, but also repeated on lines 290-291) is misleading. Figure S1 suggests that at warmer temperatures P. Syringae is most efficient. The authors can certainly describe the temperature range and the concentration range at which L01-tf-B03 becomes competitive. These conclusions would need to be revised and clarified that these statements are specifically due to a temperature range to be considered for publication.

**Response**: The reviewer raises a valid point, and we completely agree that the term 'efficiency' is misleading. The IN-activity of L01-tf-B03 may be better compared to that of *P. syringae* using the term 'potency', which we define as the concentration of INs necessary to observe IN-activity at

temperatures above the freezing point of background water (~ -23.5°C). We also now use "efficiency" to refer to the highest temperature at which an ice nucleator induces freezing, regardless of its concentration. By these definitions, *P. syringae* is more efficient since it has the highest IN-activity, but L01-tf-B03 is 1000 times more potent.

**Action taken**: We have made changes throughout the manuscript and updated the title of the paper to, "*Peltigera* lichen thalli produce highly potent ice nucleating agents". Specifically, throughout the manuscript we changed the term 'efficiency' to 'potency'. In Section 3.4, we added specific definitions on line 283 which now reads "We define potency as the concentration of INs necessary to observe IN-activity at temperatures above the freezing point of background water (~ -23.5°C). Efficiency refers to the highest temperature at which an IN induces freezing, regardless of its concentration". The updated terminology better describes the concentration range at which L01-tf-B03 "outcompetes" *P. syringae* in terms of IN-activity. We also revised the text in the Abstract and Conclusions. Line 32 now reads, "Our analysis suggests that the INs released from this fungus in culture are 1000 times more potent than the most IN-active bacterial INs from *P. syringae*" and line 331 has been updated to, "Our results show that the INs released from the fungal culture are nearly 1000 times more potent than the most IN-active bacterial cell-anchored INs classified to date".

2) The conclusion that "the ice nucleation activity of lichens does not seem to strongly correlate with the geographic region or climate zone where the samples were collected" (lines 202-203) is not supported by the presented data set. The season, storage, age of the samples likely is too variable to draw this conclusion and such a sample size.

**Response**: We appreciate the reviewer raising this point and acknowledge that we cannot make a definitive claim without obtaining IN-activity measurements from a larger sample size.

**Action taken**: We added text on line 207 that reads "A larger sample size is needed to conclusively identify whether geographic trends in IN-activity exist".

3) Handling blanks should be shown. To be clear, handling blanks represent water going through the same procedure and steps, but without the lichen. I find the reported -11 oC (line 170) to be an unacceptably high background freezing temperature. I would expect background freezing of 3 microliter droplets to be more around -25 oC. Can the authors explain and show data to support

this high background freezing? Otherwise, it's difficult to ensure that the freezing temperatures reported are due to the experimental set up vs the lichen.

**Response**: We thank the reviewer for drawing attention to this point. We recognize that this $T_{50}$ value, in the Vali-type ice nucleation assay set-up, is relatively high. The high background freezing may be due in part to dust presence, as renovations were taking place in the lab. The IN-activity of Snomax ($T_{50}$ = -3.5°C) was consistently measured as an internal control to ensure that the freezing was due to ice nucleators and not external conditions in the lab. It is also important to note that this assay is only used for a prescreening of the crude estimates of IN-activity prior to TINA measurements, which are used to validate the approximate freezing temperatures observed in the Vali-type set-up.

**Action taken**: The freezing temperature of the handling blanks was reported on line 170, but we now revised the text to clarify that the blanks were treated the same as the lichen samples. Line 170 now reads, "The IN-activity of the positive control, 1 mg mL$^{-1}$ of inactivated *P. syringae* ($T_{50}$ of -3.5°C), and water (the negative control with a $T_{50}$ of -11°C) was measured using experimental parameters identical to those of the lichen samples". We also added text on line 172 that reads, "Although the Vali-type apparatus was sufficient for crude initial tests of IN-activity, more statistically rigorous measurements were needed for quantitative analysis of the extracted INs."

4) Why were the sites chosen? Were they part of a decadal study? I have the impression that these samples were rather opportunistically chosen (rather than intentional sites) and I think this point needs to be discussed in the manuscript (it's fine if they made use of other sampling campaigns or other projects – but it should be stated explicitly). The point on different biomes is well taken and well demonstrated, but the reasoning behind why these specific sites would need to be justified.

**Response**: We appreciate the reviewer's comment. The samples were obtained opportunistically, as co-authors on this manuscript collect *Peltigera* lichens for different projects and could provide samples for our ice nucleation measurements.

**Action taken**: We updated line 94 to read, "Lichen thalli of the genus *Peltigera* were collected based on availability from prior sampling campaigns in the United States, Canada, Brazil, and Costa Rica (Fig. 1c) between February 2003 and August 2024".

5) The "N/A" values for the TINA measurements are unexplained. If the authors admit that the Vali-type temperatures are for "initial tests" (line 173), then why are there TINA measurements missing? (Same comment for Table S2 – why are TINA measurements not reported if the authors think this instrument is better?

**Response**: We thank the reviewer for this question. We chose not to measure all the lichens with TINA because the primary aim of Results 3.1 was to establish that all the tested lichens displayed IN-activity, which was evident from the combination of the two assays. The Vali-type assay, though less precise, provided sufficient evidence to identify IN-activity, and TINA was used to selectively confirm and refine these findings. Measuring all the samples with TINA would not have added more value to the manuscript beyond reiterating the IN-activity already observed with the prescreening, even though the Vali-type assay might be associated with larger error. In addition, 1 mL of sample is required for a full dilution series with TINA, and there was insufficient volume for several of the samples.

**Action taken**: We added text to line 175 in the Methods 2.4 section, which now reads, "1 mL of each investigated IN extract was serially diluted in 10-fold increments with a liquid-handling station". We also added a sentence to the Table 1 caption that reads, "Samples with sufficient volume for a full dilution series were measured with TINA".

6) The comparison between this study and their former study (2023) needs to be made clearer. Here are some suggested:

- Locations from Eufemio et al. 2023 should be included in Figure 1.
- Samples and freezing temperatures should be included in Table 1 (and see my point below about turning Table 1 into a figure)
- The authors could consider comparing quantitatively and visually the claim on lines 191-192. This comparison would be important for context.

**Response**: We thank the reviewer for raising the point about the comparison between the current study and the former one. To address the claim on line 191, which reads, "While several lichen species were previously identified as powerful INs, the prevalence of IN-activity within the genus

*Peltigera* remains largely unknown", we added a figure to the Supplement to visually compare the IN-activity of *Peltigera* lichens with other lichens.

However, it is important to clarify that the current study was not designed as a direct follow-up to the 2023 work. The earlier study provided a broad survey of IN-activity across a diverse range of lichens, including two *Peltigera*, in Alaska. In contrast, the present study focuses exclusively on *Peltigera* on a worldwide scale, including the two from Alaska. The narrower taxonomic focus and larger geographic scope of this study were intended to provide insights into the IN-activity of one of the most widespread lichen genera.

**Action taken**: We added text on line 193 to direct the reader to the Supplement and added a figure to the Supplement which shows the $T_{50}$ values of the tested *Peltigera* lichens in comparison to the four most IN-active, non-*Peltigera* lichens that were measured in the 2023 study. The introductory sentence in the Supplement reads, "Figure S1 compares the IN-activity of four lichens, *P. herrei, S. globosus, B. fuscescens,* and *C. squamosa,* measured by Eufemio et al. (2023) with the IN-activity of *Peltigera* lichens measured in the current study", and the caption reads, "Ice nucleation activity of the tested *Peltigera* lichens (represented by blue diamonds) and the four most active non-*Peltigera* lichens (*P. herrei, S. globosus, B. fuscescens,* and *C. squamosa*) measured by Eufemio et al., 2023 (shown by green squares). Shown are the $T_{50}$ values, where $T_{50}$ is defined as the temperature at which 50% of the sample droplets are frozen".

7) Heat treatments were already done in Eufemio et al. 2023 and so what is new/different about the heat treatments in this study? Data from 2023 could be included in the heat treatment Table S1 (should rather be a figure). Are heat treatments the best method for identifying proteinaceous material? I think there are multiple additional experiments (even elemental analysis) that could help further support the claim for INPs.

**Response**: We thank the reviewer for their feedback. The inclusion of heat treatments in this study was important to determine whether all *Peltigera* lichens exhibit similar responses to heat. This serves as an indicator of whether the ice-nucleating agent(s) might be proteinaceous across the diverse *Peltigera* species examined. We agree that additional (bio)chemical analyses are needed for definitive characterization of the molecular nature of lichen INs. However, the primary aims of this study were to evaluate the frequency and distribution of IN-activity in *Peltigera* and isolate

an active IN from lichen thalli. As such, molecular analyses are beyond the scope of this paper but will be important for future work.

**Action taken**: We converted Table S1 into a figure in the Supplement. The figure shows the $T_{50}$ values of the *Peltigera* lichens before and after exposure to 98°C. We rearranged and revised the final paragraph in Results 3.1 such that lines 219 – 228 now read, "All the *Peltigera* INs were found to be susceptible to heat treatments (Fig. S3 in the Supplement), supporting previous findings that lichen INs are proteinaceous, at least in part (Kieft and Ruscetti, 1990). Some species appear to be highly sensitive to heat, while others are less affected. For example, *P. aphthosa* PL729 activity is substantially lowered from -6.7°C to -9.3°C while *P. membranacea* PNW22 efficiency decreases by only 1.1°C. The differences in activation temperatures and heat sensitivity across *Peltigera* species may indicate variations in the macromolecular composition of INs. However, further experiments, including (bio)chemical analyses, are needed to decipher the molecular nature of the INs …"

8) Storage certainly plays a role in this study (the authors mention this issue only on lines 113-114). Could the authors add this information to Table 1? According to my comment below about turning Table 1 into a figure, one could add a dashed/solid outline to showcase the potential impact on freezing temperature. This point and discussion would merit a new and separate subsection and connecting to literature.

**Response**: We appreciate the reviewer's comment regarding the role of storage. We added the collection dates to Table 1. While we agree that storage conditions may influence freezing temperatures, further analysis and specific storage experiments would be required to rigorously assess these impacts, which is beyond the scope of this study. We acknowledge that we cannot exclude that sample age and storage method impacted the IN-activity but emphasize that all the tested *Peltigera* lichens were IN-active regardless.

**Action taken**: We added the years of sample collection to Table 1.  We initially attempted to create a figure to convey the information but felt that the tabular format remains the clearest way to present the findings.

9) Why do the authors think that these lichen have had so much less attention if they've been known since the 80s? (Line 55-56 and again line 82) The motivation for this study should be more

than because something is unknown. I'd be interested to hear about why the authors think P. Syringae has dominated attention. For example, in its use in Snomax.

**Response**: We thank the reviewer for their comment. While *P. syringae* is famous for its commercial applications, lichens have been comparatively overlooked despite their potential importance. This may be due in part to the ease of bacterial growth compared to lichen growth, which is often extremely slow, and the simplicity of the bacterial cell structure compared to the more complex symbiotic nature of lichens.

As noted in the manuscript, *Peltigera* is one of the most widespread lichen genera, and INs derived from lichenized fungi have been detected in the atmosphere, where they are proposed to strongly influence atmospheric processes. The presence of *Peltigera* in many parts of the world makes them particularly relevant for studying the potential sources of biological INs in the atmosphere. Furthermore, as demonstrated by the repetitive freeze-thaw cycle measurements, the stability of *Peltigera* INs suggests that these particles could remain active and exert sustained influence on atmospheric processes.

**Action taken**: We revised the text in the Conclusions, starting on line 320, to read, "A concentration series of select samples exposed to repetitive freeze-thaw revealed that *Peltigera* INs maintain activity under experimental conditions that serve as a proxy for high altitudes. Given the presence of *Peltigera* in many world regions, the lichen INs may be prevalent in airborne fungal communities where they could remain active and exert a sustained influence on atmospheric processes (Henderson-Begg et al., 2009: Moffett et al., 2015: Womack et al., 2015)".

10) The microscope data is missing and should be included in the SI. Including the images to support the claim on line 260.

**Response**: We thank the reviewer for bringing our attention to the importance of including images of cultures isolated from *P. britannica* JNU22.

**Action taken**: We added text on line 260 to direct the reader to the Supplement and updated the Supplement to include a figure showing images of three pure cultures, L01-tf-B03, L01-tf-B01, and L01-tf-A01, which were isolated from *P. britannica* JNU22 and tested for ice nucleation activity. The introductory sentence to the figure in the Supplement reads, "An optical microscope

(Zeiss Axioscope 5, Carl Zeiss Microscopy GmbH, Jena, Germany) was used to visually identify the pure cultures isolated from the thalli of *P. britannica* JNU22". The figure caption reads, "Optical microscopy at 400x magnification showing (a) L01-tf-B03, identified as a filamentous fungus, (b) L01-tf-B01, tentatively identified as a yeast, and (c) L01-tf-A01, identified as a filamentous fungus. Scale bar = 200 µm".

Specific comments:

11) Lines 50-53 mention aggregates and the first sentence is missing a reference. Consider also the paper referenced and citing (Bieber and Borduas-Dedekind, 2024; Hartmann et al., 2022; Lukas et al., 2020, 2022; Renzer et al., 2024) (most of these papers have shared authors with this manuscript, so I'm surprised they haven't been included.)

**Response**: We appreciate the reviewer pointing out these additional references.

**Action taken**: We added the suggested references starting on line 50. The text now reads, "The most active biological INs described to date are the bacterial plant pathogens, *Pseudomonas syringae*, whose ability to facilitate freezing at exceptionally warm sub-zero temperatures originates from ice nucleating proteins (INPs) located in the cell outer membrane (Govindarajan and Lindow, 1988). *P. syringae* INPs assemble into functional aggregates that are categorized into classes A-C based on freezing temperature and assembly size (Kozloff et al., 1983; Govindarajan and Lindow, 1988; Turner et al., 1990). Large protein aggregates associated with class A allow the bacteria to achieve IN-activity close to -1°C, while class C consists of comparatively small INPs active at ~ -7.5°C (Lukas et al., 2020; Lukas et al., 2022; Hartmann et al., 2022; Bieber and Borduas-Dedekind , 2024; Renzer et al., 2024).".

12) Line 66: dominate 10% of what? Lichens cover 10% of the entire surface of the earth? This point could be clarified.

**Response**: We thank the reviewer for the suggestion. We have revised the text to clarify that lichens dominate nearly 10% of earth's terrestrial ecosystems.

**Action taken**: We have amended line 66 and added Papazi et al., 2015 as an additional reference. The line now reads, "Lichens, including their fungal and prokaryotic communities (Arnold et al., 2009; Hodkinson et al., 2012), can tolerate extreme conditions and survive in environments where

other vegetation cannot, and therefore dominate nearly 10% of the earth's terrestrial ecosystems (Honegger, 2007; Papazi et al., 2015)".

13) Lines 68-69: the future reader would likely benefit from having this list of references explicitly stated. What each found and how the authors built on this previous work.

**Response**: We appreciate the reviewer's suggestion and have adapted the text to include updated references and specify that airborne reproductive propagules are a primary source of lichen-derived INs in the atmosphere.

**Action taken**:  Starting on line 68, the text now reads, "Lichen-derived INs, specifically airborne asexual reproductive propagules (Marshall, 1996; Tormo et al., 2001), have been detected in the atmosphere, where they can contribute to cloud glaciation and trigger precipitation (Henderson-Begg et al., 2009: Moffett et al., 2015)."

14) Line 94: it would be worth deleting the cardinals for the US (the directions are not listed for the other countries)

**Response**: We thank the review for pointing this out. The text has been updated accordingly.

**Action taken**: Line 94 now reads, "Lichen thalli of the genus *Peltigera* were collected based on availability from prior sampling campaigns in the United States, Canada, Brazil, and Costa Rica (Fig. 1c) between February 2003 and August 202".

15) Line 102: could the authors add to their SI the vegetation identification guides that were used?

**Response**: We thank the reviewer for their suggestion. The vegetation identification guide, by Pojar and MacKinnon (1994) is included in the manuscript references list.

**Action taken**: We updated the placement of the reference in the main text. Line 103 now reads, "Species were identified using a vegetation identification guide (Pojar and MacKinnon, 1994)…"

16) Table 1 can be converted into a figure to better visualize the breadth as well as the differences between location, symbiosis type and T50. T50 vs species in ascending/descending order of T50. (for example, using markers to represent symbiosis and colour to represent T50).

**Response**: The reviewer raises a valid point. As addressed in comment #8, we attempted to convert the table into a figure but ultimately found the tabular format to be the most informative and clear way to present the information.

17) Were some of the lands sampled from indigenous/aboriginal lands? Are there land acknowledgements to be made?

**Response**: We thank the reviewer for the thoughtful comment and appreciate the importance of land acknowledgements. The sampling in Nunavut, Canada was conducted on Indigenous lands and an acknowledgement has been included.

**Action taken**: We added the land acknowledgement to the Acknowledgment section of the manuscript.

18) Line 148: Could the authors clarify what the "hydration state of the thalli" signify?

**Response**: We thank the reviewer for drawing our attention to this definition. The hydration state of the thalli refers to the water content of the lichen specimens prior to washing and analysis. For samples stored in a frozen state, dehydration was not performed prior to freezing, so these thalli retained water upon thawing. To ensure consistency in water content across the samples, the thalli stored in a dehydrated, dormant state were rehydrated prior to washing. This step helps minimize variability in ice nucleation activity that could arise from differences in hydration.

**Action taken**: We revised the text on line 148 to read, "To ensure consistency in the water content of the thalli, specimens stored in a dehydrated, dormant state were weighed, rehydrated in pure water for 30 minutes, and reweighed prior to washing".

19) Line 167-168: how were the freezing temperatures recorded? Manually? How many replicates?

**Response**: We thank the reviewer for their comment regarding the methods. In the Vali-type assay described on line 167-168, freezing events were identified based on the optical change in appearance that is consistent with freezing, and the freezing temperatures manually recorded. Three replicates of 20 droplets each were measured. More robust measurements were later conducted with TINA.

**Action taken**: We updated the text starting on line 169 to read, "20 droplets (1 µL) of extract were cooled at 3°C min$^{-1}$ from 0°C to -20°C on a temperature-controlled aluminum plate (Linkam Scientific Instruments LTD, United Kingdom). The freezing temperature of each droplet was identified based on the optical change in appearance that occurred with freezing and the temperature at which 50% of the droplets froze, $T_{50}$, was manually recorded" and a sentence that reads, "Triplicate droplet freezing experiments were conducted" has been added on line 172.

20) Line 170: what was the positive control demonstrating? In other words, what was it controlling?

**Response**: We thank the reviewer for their comment regarding the ice nucleation assay controls. We use inactivated *P. syringae* as a positive control in our set-up since it freezes at ~ -3.5°C, significantly above background water, and serves as an example of an extremely efficient ice nucleator.

**Action taken**: As addressed above in response to comment 3#, we rearranged the text on line 170 to clarify that *P. syringae* served as the expected upper limit of IN-activity in the Vali-type set up while the freezing of the background water provided the lower limit. The text now reads, "The IN-activity of the positive control, 1 mg mL$^{-1}$ of inactivated *P. syringae* ($T_{50}$ of -3.5°C), and water (the negative control with a $T_{50}$ of -11°C) was measured using experimental parameters identical to those of the lichen samples".

21) Line 173: show data to support the claim.

**Response**: We thank the reviewer for highlighting the need to show this data. Line 173, which reads, "While the Vali-type apparatus was sufficient for initial tests of ice nucleation activity, more precise measurements were needed for quantitative analysis of the extracted INs", is supported by the procedures outlines in Methods 2.4 and Table 1, which shows the initial (Vali-type) and TINA measurements.

If the reviewer is referring to lines 170-172, which read, "Aliquots of MY medium were used as negative controls for the ice nucleation assays of the *P. britannica* JNU22 cultures and did not freeze in the investigated temperature interval", we address this point by including the freezing temperature of the medium in the text.

**Action taken**: We included the $T_{50}$ value of the prepared MY medium on line 172, which now reads, "Aliquots of MY medium were used as negative controls for the ice nucleation assays of the cultures obtained from the *P. britannica* JNU22 thallus and froze at the same $T_{50}$ value (-11°C) as the background water. We cannot definitively attribute the freezing to either the MY medium or the water".

22) Line 180: write equation explicitly.

**Response**: We thank the reviewer for the comment regarding the inclusion of Vali's equation. We have referenced Vali's original work in the manuscript to ensure proper citation and provide readers with the source for further details. While we appreciate its importance in the analysis, the equation is part of standard methodology and is comprehensively detailed in the referenced work.

23) Line 195: the term "undiluted" confused me, because the solutions needed to be made. Perhaps worth deleting? The concentration is reported further down.

**Action taken**: We removed the term "undiluted" from line 195.

24) Lines 198-199: why do the authors think that the tri- and bi-membered lichens would show difference T50 values?

**Response**: We thank the reviewer for their question. The differences in $T_{50}$ values between tri- and bimembered lichens may be influenced by differences in their composition and symbiotic structures, or by variations in the macromolecular composition of INs. However, the type of symbiosis does not have a clear impact on IN- activity, so at this time we cannot identify trends to support these ideas.

**Action taken**: As addressed on line 219, the differences in activation temperatures across *Peltigera* species may indicate variations in the macromolecular composition of INs.

25) Line 210: Key point of the paper in my opinion! This conclusion is indeed well supported in this manuscript.

**Response**: We thank the reviewer for their positive comment.

26) Line 225: Why would some samples be more sensitive to heat?

**Response:** We thank the reviewer for their question. We can speculate that differences in heat sensitivity across *Peltigera* species is due to variations in the macromolecular composition of INs. However, as addressed in the response to comment #7, further experiments are needed to adequately characterize the molecular identity of the INs.

**Action taken**: As addressed in the response to comment #7, the final paragraph of Results 3.1 now includes sentences which read, "The differences in activation temperatures and heat sensitivity across *Peltigera* species may indicate variations in the macromolecular composition of INs. However, further experiments, including chemical analyses, are needed to decipher the molecular nature of the INs…"

27) Line 318: Why was the threshold of -6.2 oC specifically chosen? Why is this temperature remarkable?

**Response**: We thank the reviewer for addressing this point. While this temperature is somewhat arbitrary, it is significant in the context of known biological ice nucleators. For instance, many fungal species, such as *Mortierella* and *Fusarium*, tend to initiate freezing at temperatures ~ -5°C to -6°C, and this range is considered high for biological ice nucleators. Setting the threshold at -6.2°C allows us to highlight *Peltigera* lichens with particularly strong IN-activity in comparison to other fungal species with notable freezing activity.

**Action taken**: We added a sentence and four references on line 318 in the Conclusions that reads, "This threshold highlights the remarkable ice-nucleating capability of *Peltigera* lichens, compared to known fungal ice nucleators (Pouleur et al., 1992; Fröhlich-Nowoisky et al., 2015; Kunert et al., 2019; Schwidetzky et al., 2023)".

28) Figure S1 which should be included in the main text. Figure 3b has a peculiar decreasing value for the x-axis. It wasn't clear to me (and likely to the future reader) what this figure was intended to highlight. Figure S1 or a figure of T50s with increasing temp could also be considered.

**Response**: We thank the reviewer for their comments regarding Fig. S1 and Fig. 3b. While the supplemental figure demonstrates that *P. syringae* has a higher initial freezing temperature, its primary purpose is to provide supplementary context. The focus of Fig. 3a is to highlight the IN-activity of L01-tf-B03, which aligns better with the main text. Regarding Fig. 3b, the intent is to

illustrate the $T_{50}$ values of L01-tf-B03 and *P. syringae* as a function of concentration. We recognize that the decreasing x-axis may have caused confusion, and we have adjusted the axis to improve clarity.

**Action taken**: We adjusted the values on the Figure 3b x-axis and updated the figure caption to improve clarity. The caption for Fig. 3b now reads, "Dilution effects on the IN-activity of L01-tf-B03 (cyan) aqueous extract and *P. syringae* in water (gray). The concentration range is 1 mg mL$^{-1}$ to 0.01 ng mL$^{-1}$. *P. syringae* INs are inactive at concentrations below 1 ng mL$^{-1}$ and are not shown."

29) More of a curiosity question, but would the authors suggest that lichens replace P. syringae in Snomax? Could be worth discussing in the implications section?

**Response**: We thank the reviewer for the insightful question. While *Peltigera* lichens demonstrate impressive IN-activity, their use as a replacement for *P. syringae* in Snomax is unlikely. Unlike bacteria, lichens are not commercially available on a large scale, and their slow growth rates make them impractical for mass production. Additionally, harvesting lichens from natural environments could pose ecological risks, as they play important roles in their ecosystems and are often sensitive to disturbance. However, if the IN of L01-tf-B03 fungal strain could be identified and produced on a large scale, it could have the potential to create a competitive product to Snomax, given the inherent instability of Snomax as a membrane-bound INP.

**Action taken**: We added a sentence to the final line of the Conclusions. Line 332 now reads, "While the slow growth and limited commercial availability of lichens make them impractical as a replacement for *P. syringae* in products such as Snomax (inactivated *P. syringae*; Snomax Int), the remarkable IN-activity of the fungal strain L01-tf-B03 at exceptionally low concentrations suggests it could be worth exploring its potential for use in commercial applications. Future research should investigate further methods to sustainably cultivate or harness these ice-nucleating components for industrial or technological purposes".